# Effect of Kaolinite and Cloisite Na+ on Storage Stability of Rubberized Binders

**DOI:** 10.3390/ma16113902

**Published:** 2023-05-23

**Authors:** Shyaamkrishnan Vigneswaran, Jihyeon Yun, Moon-Sup Lee, Soon-Jae Lee

**Affiliations:** 1Department of Engineering Technology, Texas State University, San Marcos, TX 78666, USA; gry14@txstate.edu (S.V.); yiy1@txstate.edu (J.Y.); sl31@txstate.edu (S.-J.L.); 2Korea Institute of Civil Engineering and Building Technology, Goyang-si 10223, Gyeonggi, Republic of Korea

**Keywords:** Superpave test, multiple shear creep recovery, high temperature, crumb rubber modifier, nanocomposite clay

## Abstract

This study aimed to evaluate the impact of a two-step modification process involving kaolinite and cloisite Na+ on the storage stability of rubberized binders. The process involved the manual combination of virgin binder PG 64-22 with crumb rubber modifier (CRM), which was heated to condition it. The preconditioned rubberized binder was then modified for two hours at a high speed of 8000 rpm using wet mixing. The second stage modification was performed in two parts, with part 1 using only crumb rubber as the modifier and part 2 involving the use of kaolinite and montmorillonite nano clays at a replacement percentage of 3% to the original weight of the binder along with the crumb rubber modifier. The Superpave and multiple shear creep recovery (MSCR) test methods were used to calculate the performance characteristics and separation index % of each modified binder. The results showed that the viscosity properties of kaolinite and montmorillonite improved the performance class of the binder, with montmorillonite demonstrating greater viscosity values than kaolinite even at high temperatures. Additionally, kaolinite with rubberized binders showed higher resistance to rutting, and the % recovery value from multiple shear creep recovery testing indicated that kaolinite with rubberized binders was more effective than montmorillonite with rubberized binders, even at higher load cycles. The use of kaolinite and montmorillonite reduced phase separation between the asphaltene phase and rubber-rich phase at higher temperatures, but the performance of the rubber binder was affected by higher temperatures. Overall, kaolinite with the rubber binder generally demonstrated greater binder performance.

## 1. Introduction

The rising utilization of automobiles for goods and other services has led to heightened traffic and consumption of tires, which has led to both pavement distress and piling of end-of-life tires. The pavement distress encompasses rutting, fatigue cracking, reflective cracking, and stripping [1,2,3,4,5,6]. One of the traditional methods of improving pavement wear resistance and reusing scrap tires is to use crumb rubber as an asphalt modifier, which not only increases the performance of asphalt binders but also benefits the environment by facilitating recycling [7,8,9,10]. Although crumb rubber has been the top choice as a modifier for more than 20 years due to the above advantages and extensive research studies, it can be said that the research study is still in its infancy due to the challenges such as instability when stored at elevated temperatures and when transported at elevated temperatures [11]. The problem of storage stability can be governed by Stokes’ law through a phase separation study. According to Stokes’ law
$$V_{t}=\frac{2a^{2}\Delta \rho g}{9\eta }$$
where *V*_t_ is the settling velocity of dispersed particles, *a* is the radius of dispersed particles, Δρg is the density difference between two different phases, *g* is gravitational acceleration, and *η* is the dynamic viscosity of liquid medium [12,13,14].

The current study addresses the challenge of phase separation between asphalt binder and crumb rubber due to the difference in their densities, which is governed by Stokes’ law. To mitigate this issue, smaller particle-size crumb rubber and nano-clay particles were utilized to slow down the gravitational flow of crumb rubber particles to the bottom. To ensure proper intercalation between the rubberized binders, a melt blending method was employed, which involved heating the mixture at 200 °C using a high shear mixer with a rate of 8000 rpm for 2 h to enhance homogeneity.

The nano-clay materials were selected as modifiers based on previous research with nano-clay particles in asphalt binders together with polymers, which has resulted in improved resistance to rutting, fatigue cracking, thermal cracking, and storage stability and their plentiful availability [15,16,17,18]. Moreover, previous research studies indicate that mixing nano-clay particles with asphalt binder reduces construction costs [16]. In general, nanocomposite clay particles are classified into three clay layer types 1:1, 2:1, and 2:1:1 [19,20] and only two types of nanocomposite clay materials were selected for this study: 1:1 clay-stratified kaolinite; and cloisite Na+, also known as organophilic 2:1 clay-stratified montmorillonite. The rationale for choosing these nano-clay particles is due to a plethora of research on these materials and studies showing that storage stability was improved when replacement percentages no greater than 4% by weight of the binder were used [21,22,23].

This study’s major goal is to determine how phase separation affects rubberized binders when two nanocomposite clay particles, kaolinite and organophilic montmorillonite, are used with a replacement percentage of 3% of the weight of the binder. The following tests were performed: Rotational Viscometer (RV) to find out the effect of viscosity and phase separation when exposed to two different temperatures; DSR (Dynamic Shear Rheometer) tests, measuring the modified binder on G*/Sin *δ*; and % Recovery test with MSCR when exposed to 64 °C temperature to find out its susceptibility to plastic deformation and percentage of separation. The ideal test method chosen for this study based on previous research was the cigar tube test method [24]. Figure 1 below shows a flow chart highlighting the empirical design chosen for this study.

## 2. Experimental Design

### 2.1. Materials

In this research, PG 64-22, which is a common asphalt binder used in pavement construction, was utilized as the base binder for the production of rubberized binders. The physical characteristics of the base binder, such as its viscosity and dynamic shear modulus or storage modulus, are crucial factors in determining its performance in road construction. These characteristics are provided in Table 1 for reference purposes.

The crumb rubber used to modify the binder was produced through ambient temperature grinding. This process results in a rubber material that exhibits high resistance to rutting and cracking. The use of crumb rubber as a modifier was shown to increase the viscosity of the binder, which can improve its performance in road applications. Table 2 lists the specifications of the rubber granulates used in this study, including the particle size distribution and the percentage of rubber content.

Two types of nanocomposite clay particles, namely, layered 1:1 kaolinite clay and 2:1 cloisite Na+ structure-type clay, were collected from local sources and used to modify the rubberized binders. The physical properties of these two nanocomposite clays, such as their form, molecular weight, specific gravity, and molecular formula are provided in Table 3. These nanocomposite clays were added to the rubberized binders in different proportions to evaluate their effects on the binder’s properties and performance.

To assess the storage stability of the modified binder, an aluminum cigar tube was used as a storage container, in accordance with ASTM D7173. This test method is commonly used to determine the storage stability of modified asphalt binders and can help identify any changes in the binder’s physical and chemical properties over time.

### 2.2. Production and Sampling of Nanocomposite-Modified Rubberized Binders

The wet mixing process involving a two-step modification procedure was chosen for the preparation of rubberized binders modified with nanocomposite clay particles. This process was chosen because it has been shown to result in a more homogeneously prepared and stable modified binder. The first stage modification involved mixing crumb rubber with the virgin binder PG 64-22 in weight ratios of 5% and 10% of the virgin binder by manual stirring. This was performed until a visually homogeneous mixture was obtained. This mixture was then transferred to an oven, which was maintained at a high temperature of 200 °C for a period of one hour to promote the interaction between the crumb rubber and the virgin binder.

The second stage modification involved mixing the nanocomposite clay particles separately with the 5% and 10% rubberized binders for two hours using a high-shear mixer at a speed of 8000 rpm. This was performed to ensure that the nanocomposite clay particles were uniformly distributed in the binder, which can improve its mechanical properties and resistance to deformation. After modification, samples were immediately taken to check the viscosity and viscoelastic properties to ensure that the modified binder met the desired specifications.

The modified binder residues were then poured into a 50 ± 0.5 g aluminum cigar tube, which was held vertically and sealed at the top. This tube was then placed in the oven at 163 ± 5 °C for 48 ± 1 h for conditioning. This high-temperature conditioning was necessary to further enhance the interaction between the crumb rubber, nanocomposite clay particles, and the virgin binder and to ensure that the modified binder was stable and had the desired properties. After conditioning, the samples were transferred to a refrigerator maintained at −10 ± 10 °C for at least 4 h until the molten binder solidified. The solidified aluminum cigar tube was then divided into three equal halves and placed in the oven at 163 ± 5 °C until the binding agent began to melt. The modified binder was then further liquefied in a hotplate and stirred manually with a spatula free of foreign substances. Samples were taken to check the viscosity and viscoelastic properties within 30 min of the sampling process to avoid oxidation reactions.

The choice of the aluminum cigar tube as a container for storage stability evaluation was based on its non-reactive nature and its ability to maintain the modified binder’s properties over time. Overall, this modified binder preparation procedure was chosen to ensure that the modified binder was homogeneously prepared, had stable viscoelastic properties, and met the desired specifications for use in asphalt pavement applications.

### 2.3. Evaluation of Binder

#### 2.3.1. Rotational Viscosity

This is a test method used to evaluate the mix workability factor from the start of asphalt production at the plant to compaction of the asphalt mix in the field. Viscosity properties were evaluated using a Brookfield rotational viscometer and tests were performed according to AASHTO T 316. Spindle #27 with a sample size of 10.5 g was used for this study. Spindle rpm was maintained at 20 and a total of 20 readings were taken at 1 min intervals.

#### 2.3.2. Rheological Properties and MSCR

In this study, a dynamic shear rheometer (DSR) was utilized to analyze the viscoelastic properties of the modified binder. Two properties were considered for this study using this device: (i) G*/Sin *δ* per ASTM D7175, a permanent deformation study, and (ii) % recovery at both 0.1 kPa and 3.2 kPa per ASTM D7405. Both tests were performed on the same sample at the same temperature of 64 °C.

#### 2.3.3. Separation Index/Phase Separation

Only the top and bottom parts of the aluminum cigar case were considered for the separation index study. First, a high-temperature viscosity property was used to calculate the SI based on Equation (1), where (viscosity)max is the higher value between the top and bottom parts and (viscosity)avg is the average value for both parts. According to the above calculation for SI, G*/Sin *δ* was used to evaluate the separation index ratio via Equation (2) [25,26,27,28,29,30,31,32,33,34]. Finally, the two results of the SI were compared.
(1)$$\mathrm{Separation}\;\mathrm{index}=\frac{\left(Viscosity\right)max-\left(Viscosity\right)avg}{\left(\left(Viscosity\right)avg\right)}$$
(2)$$\mathrm{Separation}\;\mathrm{index}=\frac{\left(G*/Sin\delta \right)max-\left(G*/Sin\delta \right)avg}{\left(\left(G*/Sin\delta \right)avg\right)}$$

## 3. Results and Discussions

### 3.1. Rotational Viscosity

Asphalt binder viscosity is an important factor affecting workability during the manufacture, delivery, and compaction of asphalt mixes. If the binder viscosity is too high, it can be challenging to achieve the optimum density in the field, which is also linked to the life of the pavement. To understand the effect of different modifiers on the viscosity of asphalt binders, a study was conducted on modified binders using crumb rubber, kaolinite clay, and Cloisite Na+ (organophilic montmorillonite).

The viscosity values at 135 °C for the modified binder were measured immediately after mixing and after conditioning the binder in the oven for 48 ± 1 h. As expected, the binder viscosity increased with an increase in the content of crumb rubber modifier (CRM) in the binder. The modified binder with nanocomposite clay and CRM showed a higher viscosity compared to the binder modified with only crumb rubber. From Figure 2, it is clear that the modified binder is workable immediately after modification, as the viscosity values were found to be below 3000 cp, indicating that field compaction can be achieved.

However, after conditioning the binder in the oven for 48 ± 1 h, the lower portion of the sample was found to be unworkable except for the binder modified with only 5% CRM content. The chart’s overall trend demonstrated that at 135 °C, increasing the crumb rubber percentage to 10% from 5% with 3% kaolinite clay showed a decreasing trend in viscosity in the lower half but an opposing trend in the original, upper, and middle portions. This clearly indicates that using kaolinite with crumb rubber particles will result in an unstable mix.

In contrast, the use of Cloisite Na+ showed a positive trend, as the percentage of rubber crumb particles with montmorillonite increased from 5% to 10%, resulting in an increase in viscosity. When the percentage of crumb rubber substitute increased to 10% with 3% montmorillonite compared to 10% crumb rubber and 3% kaolinite, the viscosity was almost 12,000 cp, which would make it almost impossible to lay and compact the mix in the field. However, as the temperature increases from 135 to 180 °C, the viscosity values start to decrease, as seen in Figure 3. Similar to 135 °C, the kaolinite and crumb-rubber-modified binder had insignificant viscosity levels, and montmorillonite-modified rubberized binders showed an increasing trend as crumb rubber replacement percentages increased.

Using a one-way ANOVA analysis of variance, the statistical significance of the modified binder viscosity values for 135 °C and 180 °C, accounting for the original, top, and bottom parts, was examined with a 95% confidence interval and is presented below in Table 4 and Table 5. The statistical analysis results were similar at both temperatures. When comparing the sample mean value within the same population, the values were found to be significant in both the pristine and post-conditioning states of the lower portion, except when comparing kaolinite-added rubberized binders. In the upper part, the values for both 10% crumb rubber and 5% crumb rubber with 3% kaolinite were found to be insignificant because increasing the percentage of rubber substitute results in an increase in viscosity, but using kaolinite with crumb rubber had the opposite effect, which is the reason for the insignificance. These findings provide valuable insights into the use of different modifiers in asphalt binder to improve its workability and ultimately the quality of the pavement.

### 3.2. Rheological Properties

This study focused on examining the permanent deformation property of binder by conducting original binder tests at 64 °C using a Dynamic Shear Rheometer (DSR). The G*/Sin *δ* values were determined for both the original condition and after conditioning of the samples. The conditioned samples were further divided into three parts and tested, and G*/Sin *δ* values were determined. The results of the study were found to be consistent with the viscosity values, with the exception of the kaolinite-modified rubberized binder. This binder showed an increase in G*/Sin *δ* values with the addition of kaolinite clay and crumb rubber content, in contrast to the viscosity test results. The lower portion of the sample showed a similar trend in G*/Sin *δ* values before and after conditioning, whereas the top and middle portions showed a decrease, as seen in Figure 4, indicating the movement of rubber crumb particles downwards due to gravity. The kaolinite-clay-modified binder was found to have higher G*/Sin *δ* values than the montmorillonite-clay-modified rubberized binder, indicating that rubberized binders modified with kaolinite clay have a higher resistance to rutting than those modified with montmorillonite clay.

Statistical analysis was performed using a one-way ANOVA with a 95% confidence interval, similar to the viscosity test. The statistical significance of the change in different contents was examined by comparing the original condition and the top and bottom parts of the sample after conditioning (see Table 6). The results showed statistical significance, except for a few parts of the top section, where the G*/Sin *δ* values decreased compared to the original condition due to the movement of rubber particles toward the bottom of the tube. Moreover, the significant difference within the same population means was limited to a single temperature of 64 °C. These findings provide strong scientific evidence that the addition of kaolinite clay and crumb rubber content improves the resistance to rutting of rubberized binders, particularly when compared to those modified with montmorillonite clay.

### 3.3. Multi-Stress Creep and Recovery Property Test

The Multi-Stress Creep and Recovery Test (MSCR) is a widely used method to evaluate the rheological properties of asphalt binders. The MSCR test is crucial in predicting the binder’s resistance to rutting and cracking under various loads, including elevated temperature and humidity, as well as its ability to recover from these stresses. The MSCR test was performed at 64 °C following AASHTO TP70 guidelines, with the sample loaded at both 0.1 kPa and 3.2 kPa, to assess the viscoelasticity of the modified binders under more severe conditions than those encountered in the Dynamic Shear Rheometer (DSR) test.

Figure 5 and Figure 6 illustrate the percentage recovery values at 0.1 kPa and 3.2 kPa, respectively. The results show that as the level of CRM increased, the percent recovery of the modified binder in the bottom of the conditioned binder increased. However, after conditioning the 3% montmorillonite with 5% rubber particles, a higher recovery percentage was observed than that of 3% kaolinite with 5% rubber particles. The top and middle portions of the sample showed very little recovery percentage at both 0.1 kPa and 3.2 kPa, primarily due to the deposition of the rubber crumb particles on the bottom tube. Moreover, both nanocomposite clay particles had almost similar percentage recovery values, with montmorillonite performing better than kaolinite at 5% crumb rubber content, and the opposite observed at 10% crumb rubber.

The percentage recovery values serve as an indicator of the binder’s resistance to rutting. The higher the percentage recovery, the higher the binder’s resistance to rutting. Both kaolinite and montmorillonite exhibited resistance to rutting, but at a higher percentage replacement with crumb rubber, kaolinite performed better at both loads.

Statistical analyses were performed using a one-way ANOVA and a 95% confidence interval to determine the statistical significance of the changes in percentage recovery between different contents of the modified binder. The statistical analysis was performed at both 0.1 kPa and 3.2 kPa, with the temperature maintained at 64 °C (Table 7 and Table 8).

Comparing the sample means between the same population, including original–original, original–top, original–bottom, and similar top–top, top–bottom, etc., using a one-way ANOVA, it was observed that 5% crumb rubber with 3% kaolinite and 5% crumb rubber with 3% montmorillonite were insignificant when comparing the original–original population means. For the remainder of the comparison, significant differences were observed. The movement of the crumb rubber particles to the bottom of the cigar tube resulted in several insignificant values when comparing top–top population means.

Comparing bottom to bottom at a lower load of 0.1 kPa, there was not a large significant difference, with similar values observed. However, when the load increased to 3.2 kPa, there was a significant difference between the population means, indicating a lower percentage recovery and lower resistance to rutting. This observation highlights the fact that the higher the load, the greater the effect of the ruts, and the reduced service life of the pavement.

### 3.4. Storage Stability Results

In order to assess the storage stability of modified binders, the rheological properties of G*/Sin *δ* and viscosity of top and bottom samples were utilized to calculate the Superpave proposed Separation Index (SI). The phase separation values for different modifiers at varying temperatures are presented in Table 9. The data indicate that the percentage of separation increases with a higher content of crumb rubber and mixing temperature, but only when crumb rubber is used as the sole modifier. The inclusion of stabilizers, such as nanocomposite clay, had an inverse effect as the temperature increased; the phase separation between the asphaltene phase and rubber phase decreased. The use of 3% kaolinite had an inverse effect on the crumb rubber content, as the percentage of crumb rubber increasing from 5% to 10% reduced the separation between the two layers. It is evident from Table 9 that the use of stabilizers had a positive effect at a higher percentage replacement of crumb rubber, but an inverse effect of increasing the percentage of separation when replacement percentages were used at a lower level of 5%. Among the two nanocomposite clays, kaolinite had a lower percentage separation of 44% compared to montmorillonite at 69%.

The study of the G*/Sin *δ* phase separation was limited to a temperature of 64 °C. The results from Table 10 indicate that both stabilizers had no effect on reducing the phase separation between the asphaltene phase and rubber phase. The viscosity study demonstrates that stabilizers only work better at elevated temperatures. It is crucial to maintain the storage stability of modified binders, as the separation of the rubber phase can lead to the degradation of the asphalt binder. This, in turn, can result in reduced durability and performance of the pavement, ultimately leading to premature failure and costly repairs. Therefore, understanding the effects of different modifiers on the storage stability of asphalt binders is essential for designing long-lasting and cost-effective pavement structures.

## 4. Summary and Conclusions

The storage stability of modified binders containing 5% and 10% rubber crumb modified separately with 3% kaolinite and 3% cloisite Na+ stabilizers was investigated. First, the binders were mixed manually with different amounts of rubber crumbs and conditioned at 200 °C for one hour. Later, the oven-conditioned binder was subjected to high shear mixing at 8000 rpm and the samples were conditioned in an oven at 163 °C for 48 ± 1 h and then cooled to solidify the material. The properties and phase separation of the modified product were determined using a rotational viscometer and a dynamic shear rheometer immediately after modification and conditioning.

The addition of crumb rubber increased the viscosity of the binder, with conditioned binders showing higher viscosity in the lower part compared to the upper and middle parts due to the crumb rubber particles moving down from the top part. The use of stabilizers such as kaolinite and cloisite Na+ had a direct impact on viscosity, with both nanocomposite clay particles increasing the viscosity of the mixture and having an inverse effect on viscosity as temperature increased. The use of montmorillonite increased the viscosity of the mix at a higher percentage of rubber crumb substitute.

The DSR test at a temperature of 64 °C showed that the use of the two nanocomposite clay particles increased the G*/Sin *δ* in the native state with increasing rubber content. The modified binders after conditioning showed the highest G*/Sin *δ* value in the lower part, with kaolinite clay with crumb rubber performing better than montmorillonite, indicating that the use of kaolinite clay with crumb rubber blended binder increases rut resistance.

The MSCR test showed that the percentage recovery decreased as loads increased, with rubberized kaolinite-clay-modified binder showing a higher percent recovery at a higher replacement percentage of crumb rubber, while Cloisite Na+ had a higher percent recovery at a lower percent replacement of crumb rubber. Both nanocomposite-clay-modified rubberized binders improved pavement performance and pavement life.

The SI% of G*/Sin *δ* generally increased with the percentage of crumb rubber content and addition of nanocomposite clay content in the binder. The G*/Sin *δ* results showed that the addition of stabilizers such as nanocomposite clay particles did not help reduce the phase separation between the asphaltene phase and the rubber-rich phase since the temperature was limited to 64 °C. The SI% from viscosity tests were useful for evaluating the storage stability of modified binders compared to the SI% from G*/Sin *δ* using DSR, as higher temperature helped reduce phase separation when stabilizers were mixed with the rubberized binders. The modification of 3% montmorillonite with rubberized binders followed a clear pattern in both the Superpave test and the MSCR test, while the use of kaolinite as a stabilizer had a significant effect on improving binder performance, but each test followed the other with marked differences in a pattern that clearly points to montmorillonite as a more stable material.

The results were limited to two levels of rubber crumb, two types of nanocomposite clay, and PG 64-22 virgin binder, and were intended to show the variation in storage stability according to binder levels. To draw a more general conclusion, it is recommended to use distinct types of rubber crumb, nanocomposite clay, and asphalt binder. In addition, for future studies, a study evaluation of how the SI changes as a function of longer mixing time at high temperature can be considered. The results of this study provide insights into the effects of stabilizers and crumb rubber on the storage stability of modified binders.

## Figures and Tables

**Figure 1 materials-16-03902-f001:**
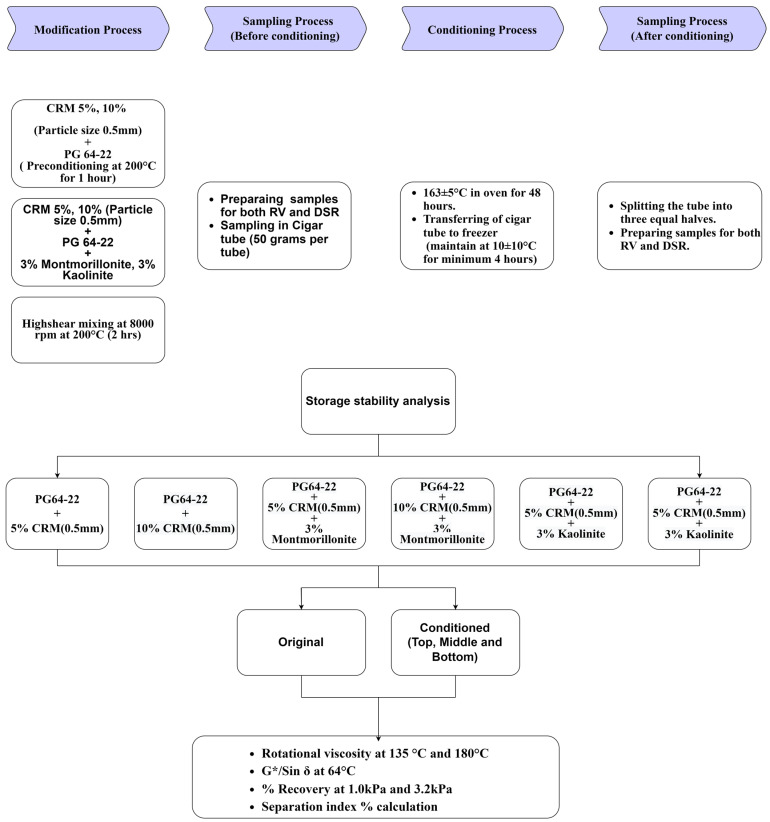
Flow chart of experimental design procedures.

**Figure 2 materials-16-03902-f002:**
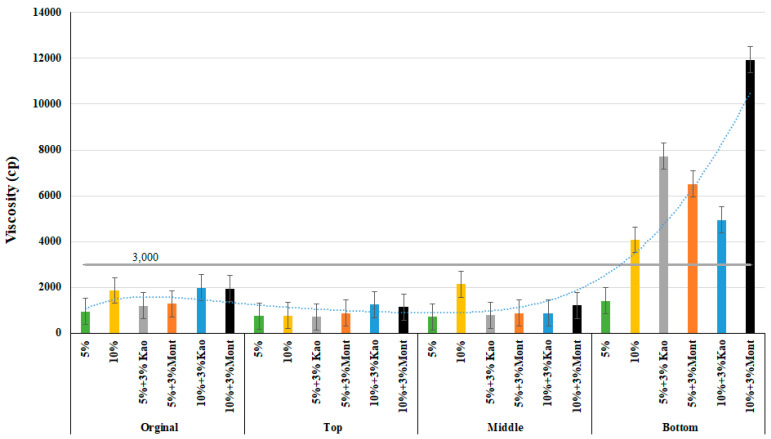
Viscosity at 135 °C of the modified binders of original, top, middle, and bottom parts.

**Figure 3 materials-16-03902-f003:**
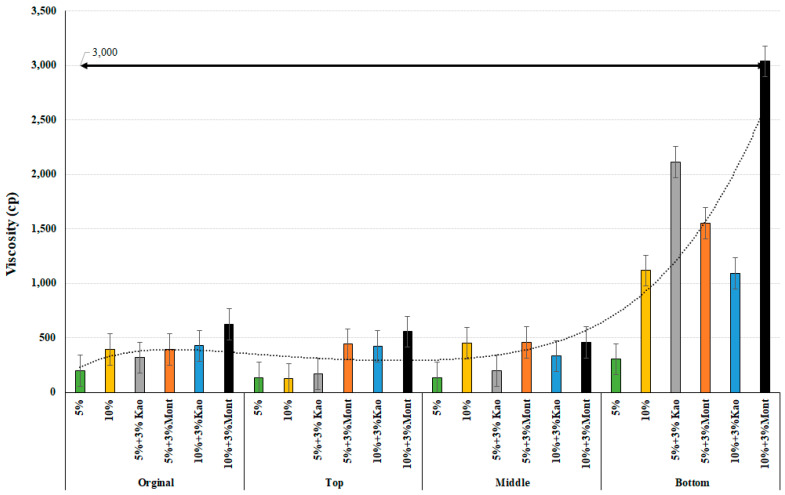
Viscosity at 180 °C of the modified binders of original, top, middle, and bottom parts.

**Figure 4 materials-16-03902-f004:**
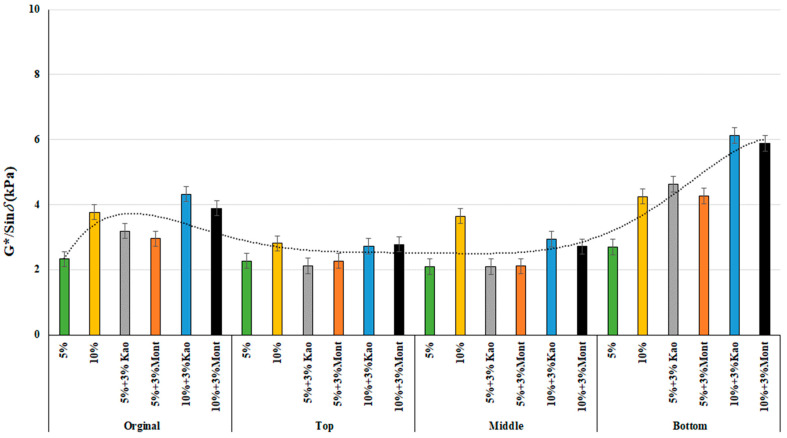
G*/sin δ at 64 °C of the modified binders of original, top, middle, and bottom parts.

**Figure 5 materials-16-03902-f005:**
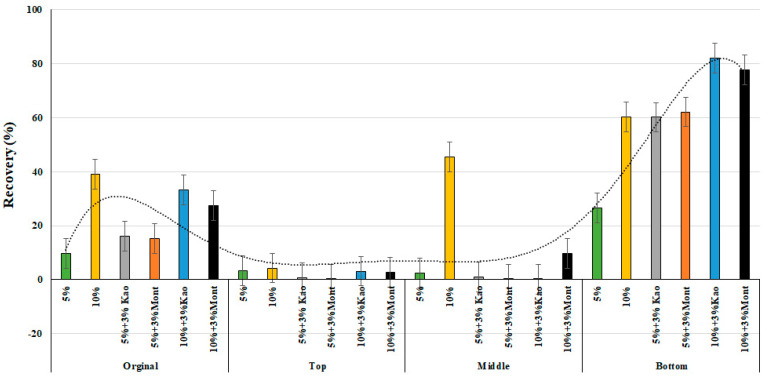
% Recovery of 0.1 kPa at 64 °C of the modified asphalt binders of original, top, middle, and bottom parts.

**Figure 6 materials-16-03902-f006:**
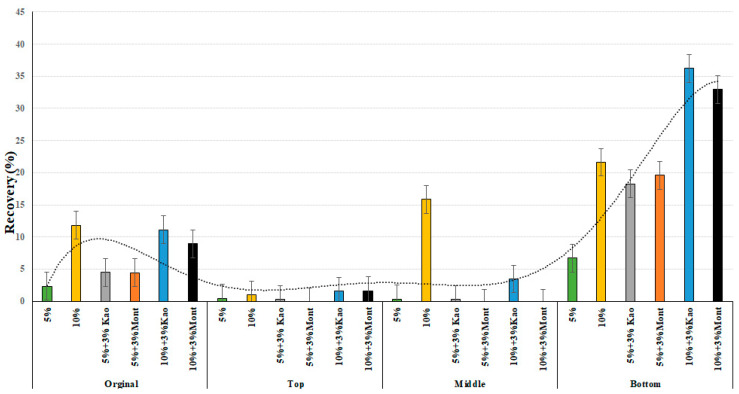
% Recovery of 3.2 kPa at 64 °C of the modified asphalt binders of original, top, middle, and bottom parts.

**Table 1 materials-16-03902-t001:** Properties of virgin asphalt binder (PG 64-22).

Aging States	Test Properties	Test Result
Unaged binder	Viscosity @ 135 °C (cP)	537
	G*/sin *δ* @ 64 °C (kPa)	1.40
RTFO aged residual	G*/sin *δ* @ 64 °C (kPa)	3.84
RTFO + PAV aged residual	G*sin *δ* @ 25 °C (kPa)	4403
	Stiffness @ −12 °C (MPa)	206
	m-value @ −12 °C	0.324

**Table 2 materials-16-03902-t002:** Gradation of crumb rubber used in this study.

Sieve Number (µm)	% Cumulative Passed Size ≤ 0.5 mm
4	100
8	100
30	100
50	58.7
100	14.7
200	2.0

**Table 3 materials-16-03902-t003:** Physical properties of nanocomposite clay.

Materials	Molecular Formula	Molecular Weight	Form	Specific Gravity
Kaolinite clay	Al_2_Si_2_O_5_(OH)_4_	258.2 g/mol	Powder	~2.6
Cloisite Na+	Na_0.7_(Mg_6.5_Li_0.3_Si_8_O_20_) (OH)_4_(OH_2_)_4_Na_0.7_ + 2.3H_2_O	1190 (g/mol)	Powder	~2.8

**Table 4 materials-16-03902-t004:** Statistical analysis results of the viscosity at 135 °C of modified binders as a function of original, top, and bottom parts (α = 0.05).

		Original						Top						Bottom					
	Viscosity (135 °C)	5%	10%	5% + 3% Kao	5% + 3% Mon	10% + 3% Kao	10% + 3% Mon	5%	10%	5% + 3% Kao	5% + 3% Mon	10% + 3% Kao	10% + 3% Mon	5%	10%	5% + 3% Kao	5% + 3% Mon	10% + 3% Kao	10% + 3% Mon
Original	5%	-	S	S	S	S	S	S	S	S	N	S	S	S	S	S	S	S	S
	10%	-	-	S	S	S	S	S	S	S	S	S	S	S	S	S	S	S	S
	5% + 3% Kao	-	-	-	S	S	S	S	S	S	S	N	N	S	S	S	S	S	S
	5% + 3% Mon	-	-	-	-	S	S	S	S	S	S	N	S	S	S	S	S	S	S
	10% + 3% Kao	-	-	-	-	-	N	S	S	S	S	S	S	S	S	S	S	S	S
	10% + 3% Mon	-	-	-	-	-	-	S	S	S	S	S	S	S	S	S	S	S	S
Top (%)	5%	-	-	-	-	-	-	-	N	N	S	S	S	S	S	S	S	S	S
	10%	-	-	-	-	-	-	-	-	N	S	S	S	S	S	S	S	S	S
	5% + 3% Kao	-	-	-	-	-	-	-	-	-	S	S	S	S	S	S	S	S	S
	5% + 3% Mon	-	-	-	-	-	-	-	-	-	-	S	S	S	S	S	S	S	S
	10% + 3% Kao	-	-	-	-	-	-	-	-	-	-	-	S	S	S	S	S	S	S
	10% + 3% Mon	-	-	-	-	-	-	-	-	-	-	-	-	S	S	S	S	S	S
Bottom (%)	5%	-	-	-	-	-	-	-	-	-	-	-	-	-	S	S	S	S	S
	10%	-	-	-	-	-	-	-	-	-	-	-	-	-	-	S	S	S	S
	5% + 3% Kao	-	-	-	-	-	-	-	-	-	-	-	-	-	-	-	S	S	S
	5% + 3% Mon	-	-	-	-	-	-	-	-	-	-	-	-	-	-	-	-	S	S
	10% + 3% Kao	-	-	-	-	-	-	-	-	-	-	-	-	-	-	-	-	-	S
	10% + 3% Mon	-	-	-	-	-	-	-	-	-	-	-	-	-	-	-	-	-	-

S—Significant. N—Non-significant.

**Table 5 materials-16-03902-t005:** Statistical analysis results of the viscosity at 180 °C of modified binders as a function of original, top, and bottom parts (α = 0.05).

		Original						Top						Bottom					
	Viscosity (180 °C)	5%	10%	5% + 3% Kao	5% + 3% Mon	10% + 3% Kao	10% + 3% Mon	5%	10%	5% + 3% Kao	5% + 3% Mon	10% + 3% Kao	10% + 3% Mon	5%	10%	5% + 3% Kao	5% + 3% Mon	10% + 3% Kao	10% + 3% Mon
Original	5%	-	S	S	S	S	S	S	S	N	S	S	S	S	S	S	S	S	S
	10%	-	-	S	N	N	S	S	S	S	S	N	S	S	S	S	S	S	S
	5% + 3% Kao	-	-	-	S	S	S	S	S	S	S	S	S	N	S	S	S	S	S
	5% + 3% Mon	-	-	-	-	N	S	S	S	S	S	N	S	S	S	S	S	S	S
	10% + 3% Kao	-	-	-	-	-	S	S	S	S	N	N	S	S	S	S	S	S	S
	10% + 3% Mon	-	-	-	-	-	-	S	S	S	S	S	S	S	S	S	S	S	S
Top (%)	5%	-	-	-	-	-	-	-	N	N	S	S	S	S	S	S	S	S	S
	10%	-	-	-	-	-	-	-	-	S	S	S	S	S	S	S	S	S	S
	5% + 3% Kao	-	-	-	-	-	-	-	-	-	S	S	S	S	S	S	S	S	S
	5% + 3% Mon	-	-	-	-	-	-	-	-	-	-	N	S	S	S	S	S	S	S
	10% + 3% Kao	-	-	-	-	-	-	-	-	-	-	-	S	S	S	S	S	S	S
	10% + 3% Mon	-	-	-	-	-	-	-	-	-	-	-	-	S	S	S	S	S	S
Bottom (%)	5%	-	-	-	-	-	-	-	-	-	-	-	-	-	S	S	S	S	S
	10%	-	-	-	-	-	-	-	-	-	-	-	-	-	-	S	S	N	S
	5% + 3% Kao	-	-	-	-	-	-	-	-	-	-	-	-	-	-	-	S	S	S
	5% + 3% Mon	-	-	-	-	-	-	-	-	-	-	-	-	-	-	-	-	S	S
	10% + 3% Kao	-	-	-	-	-	-	-	-	-	-	-	-	-	-	-	-	-	S
	10% + 3% Mon	-	-	-	-	-	-	-	-	-	-	-	-	-	-	-	-	-	-

S—Significant. N—Non-significant.

**Table 6 materials-16-03902-t006:** Statistical analysis results of the G*/Sin *δ* at 64 °C of modified binders as a function of original, top, and bottom parts (α = 0.05).

		Original						Top						Bottom					
	G*/Sin *δ* (64 °C)	5%	10%	5% + 3% Kao	5% + 3% Mon	10% + 3% Kao	10% + 3% Mon	5%	10%	5% + 3% Kao	5% + 3% Mon	10% + 3% Kao	10% + 3% Mon	5%	10%	5% + 3% Kao	5% + 3% Mon	10% + 3% Kao	10% + 3% Mon
Original	5%	-	S	S	S	S	S	N	S	S	N	S	S	S	S	S	S	S	S
	10%	-	-	S	S	S	S	S	S	S	S	S	S	S	S	S	S	S	S
	5% + 3% Kao	-	-	-	S	S	S	S	S	S	S	S	S	S	S	S	S	S	S
	5% + 3% Mon	-	-	-	-	S	S	S	S	S	S	S	S	S	S	S	S	S	S
	10% + 3% Kao	-	-	-	-	-	S	S	S	S	S	S	S	S	S	S	N	S	S
	10% + 3% Mon	-	-	-	-	-	-	S	S	S	S	S	S	S	S	S	S	S	S
Top (%)	5%	-	-	-	-	-	-	-	S	S	N	S	S	S	S	S	S	S	S
	10%	-	-	-	-	-	-	-	-	S	S	S	N	S	S	S	S	S	S
	5% + 3% Kao	-	-	-	-	-	-	-	-	-	S	S	S	S	S	S	S	S	S
	5% + 3% Mon	-	-	-	-	-	-	-	-	-	-	S	S	S	S	S	S	S	S
	10% + 3% Kao	-	-	-	-	-	-	-	-	-	-	-	N	N	S	S	S	S	S
	10% + 3% Mon	-	-	-	-	-	-	-	-	-	-	-	-	N	S	S	S	S	S
Bottom (%)	5%	-	-	-	-	-	-	-	-	-	-	-	-	-	S	S	S	S	S
	10%	-	-	-	-	-	-	-	-	-	-	-	-	-	-	S	N	S	S
	5% + 3% Kao	-	-	-	-	-	-	-	-	-	-	-	-	-	-	-	S	S	S
	5% + 3% Mon	-	-	-	-	-	-	-	-	-	-	-	-	-	-	-	-	S	S
	10% + 3% Kao	-	-	-	-	-	-	-	-	-	-	-	-	-	-	-	-	-	S
	10% + 3% Mon	-	-	-	-	-	-	-	-	-	-	-	-	-	-	-	-	-	-

S—Significant. N—Non-significant.

**Table 7 materials-16-03902-t007:** Statistical analysis results of the % recovery of 0.1 kPa at 64 °C of modified binders as a function of original, top, and bottom parts (α = 0.05).

		Original						Top						Bottom					
	% Recovery 0.1 kPa (64 °C)	5%	10%	5% + 3% Kao	5% + 3% Mon	10% + 3% Kao	10% + 3% Mon	5%	10%	5% + 3% Kao	5% + 3% Mon	10% + 3% Kao	10% + 3% Mon	5%	10%	5% + 3% Kao	5% + 3% Mon	10% + 3% Kao	10% + 3% Mon
Original	5%	-	S	S	S	S	S	S	S	S	S	S	S	S	S	S	S	S	S
	10%	-	-	S	S	S	S	S	S	S	S	S	S	S	S	S	S	S	S
	5% + 3% Kao	-	-	-	N	S	S	S	S	S	S	S	S	S	S	S	S	S	S
	5% + 3% Mon	-	-	-	-	S	S	S	S	S	S	S	S	S	S	S	S	S	S
	10% + 3% Kao	-	-	-	-	-	S	S	S	S	S	S	S	S	S	S	S	S	S
	10% + 3% Mon	-	-	-	-	-	-	S	S	S	S	S	S	N	S	S	S	S	S
Top (%)	5%	-	-	-	-	-	-	-	N	S	S	N	N	S	S	S	S	S	S
	10%	-	-	-	-	-	-	-	-	S	S	N	N	S	S	S	S	S	S
	5% + 3% Kao	-	-	-	-	-	-	-	-	-	N	N	N	S	S	S	S	S	S
	5% + 3% Mon	-	-	-	-	-	-	-	-	-	-	S	S	S	S	S	S	S	S
	10% + 3% Kao	-	-	-	-	-	-	-	-	-	-	-	N	S	S	S	S	S	S
	10% + 3% Mon	-	-	-	-	-	-	-	-	-	-	-	-	S	S	S	S	S	S
Bottom (%)	5%	-	-	-	-	-	-	-	-	-	-	-	-	-	S	S	S	S	S
	10%	-	-	-	-	-	-	-	-	-	-	-	-	-	-	N	N	S	S
	5% + 3% Kao	-	-	-	-	-	-	-	-	-	-	-	-	-	-	-	N	S	S
	5% + 3% Mon	-	-	-	-	-	-	-	-	-	-	-	-	-	-	-	-	S	S
	10% + 3% Kao	-	-	-	-	-	-	-	-	-	-	-	-	-	-	-	-	-	S
	10% + 3% Mon	-	-	-	-	-	-	-	-	-	-	-	-	-	-	-	-	-	-

S—Significant. N—Non-significant.

**Table 8 materials-16-03902-t008:** Statistical analysis results of the % rec of 3.2 kPa at 64 °C of modified binders as a function of original, top, and bottom parts (α = 0.05).

		Original						Top						Bottom					
	% Recovery 0.1 kPa (64 °C)	5%	10%	5% + 3% Kao	5% + 3% Mon	10% + 3% Kao	10% + 3% Mon	5%	10%	5% + 3% Kao	5% + 3% Mon	10% + 3% Kao	10% + 3% Mon	5%	10%	5% + 3% Kao	5% + 3% Mon	10% + 3% Kao	10% + 3% Mon
Original	5%	-	S	S	S	S	S	S	S	S	S	S	S	S	S	S	S	S	S
	10%	-	-	S	S	S	S	S	S	S	S	S	S	S	S	S	S	S	S
	5% + 3% Kao	-	-	-	N	S	S	S	S	S	S	S	S	S	S	S	S	S	S
	5% + 3% Mon	-	-	-	-	S	S	S	S	S	S	S	S	S	S	S	S	S	S
	10% + 3% Kao	-	-	-	-	-	S	S	S	S	S	S	S	S	S	S	S	S	S
	10% + 3% Mon	-	-	-	-	-	-	S	S	S	S	S	S	S	S	S	S	S	S
Top (%)	5%	-	-	-	-	-	-	-	S	N	S	S	S	S	S	S	S	S	S
	10%	-	-	-	-	-	-	-	-	S	S	S	S	S	S	S	S	S	S
	5% + 3% Kao	-	-	-	-	-	-	-	-	-	N	S	S	S	S	S	S	S	S
	5% + 3% Mon	-	-	-	-	-	-	-	-	-	-	S	S	S	S	S	S	S	S
	10% + 3% Kao	-	-	-	-	-	-	-	-	-	-	-	N	S	S	S	S	S	S
	10% + 3% Mon	-	-	-	-	-	-	-	-	-	-	-	-	S	S	S	S	S	S
Bottom (%)	5%	-	-	-	-	-	-	-	-	-	-	-	-	-	S	S	S	S	S
	10%	-	-	-	-	-	-	-	-	-	-	-	-	-	-	S	S	S	S
	5% + 3% Kao	-	-	-	-	-	-	-	-	-	-	-	-	-	-	-	S	S	S
	5% + 3% Mon	-	-	-	-	-	-	-	-	-	-	-	-	-	-	-	-	S	S
	10% + 3% Kao	-	-	-	-	-	-	-	-	-	-	-	-	-	-	-	-	-	S
	10% + 3% Mon	-	-	-	-	-	-	-	-	-	-	-	-	-	-	-	-	-	-

S—Significant. N—Non-significant.

**Table 9 materials-16-03902-t009:** Separation index from viscosity of modified binders.

Temperature	Binder	Viscosity (cp)		
		Top	Bottom	% Separation
135 C	CRM 5%	742.5	1405	31
	CRM 10%	762.5	4075	68
	CRM 5% + 3% Kao	712.5	7725	83
	CRM 5% + 3% Mon	868	6518.8	76
	CRM 10% + 3% Kao	1225	4887.5	60
	CRM 10% + 3% Mon	1150	11931	82
180 C	CRM 5%	136.3	305	38
	CRM 10%	125	1118.8	80
	CRM 5% + 3% Kao	172.5	2112.5	85
	CRM 5% + 3% Mon	435	1555	56
	CRM 10% + 3% Kao	420	1092.5	44
	CRM 10% + 3% Mon	551	3039	69

**Table 10 materials-16-03902-t010:** Separation index from G*/sin *δ* of modified binders.

Temperature	Binder	G*/Sin *δ* (kPa)		
		Top	Bottom	% Separation
64 C	CRM 5%	2.27	2.7	9
	CRM 10%	2.81	3.65	20
	CRM 5% + 3% Kao	2.12	4.63	37
	CRM 5% + 3% Mon	2.28	4.27	30
	CRM 10% + 3% Kao	2.72	6.13	39
	CRM 10% + 3% Mon	2.78	5.89	36

## Data Availability

The data used to support the findings of this study are included within the article.

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
