# Peer review of "Effect of Kaolinite and Cloisite Na+ on Storage Stability of Rubberized Binders"

_materials, 2023, doi:10.3390/ma16113902_

Round 1

Reviewer 1 Report

  • This work investigated the storage stability of the modified binders containing 5% and 10% modified separately with crumb rubber, and both portions of rubberized binders modified separately with stabilizer 3% kaolinite and 3% cloisite Na+.

  • Please note the formality of the format. 

  • Please modify the unit in Table 9 and Tabe 10.

  •  

Quality of English language need improvement

Reviewer 2 Report

The manuscript "Effect of Kaolinite and Cloisite Na+ on Storage Stability of Rubberized Binders" investigate the influence of kaolinite and cloisite Na+ on the storage stability of rubberized binders by a two-step modification process.

The work " "Effect of Kaolinite and Cloisite Na+ on Storage Stability of Rubberized Binders" has an actual and interesting subject of the research field.

The work is of good value and can be considered for publication.

The presented results are useful, but unfortunately in every part of the manuscript the authors have to intervene to bring the work to the standards of this journal.

The REFERENCES section must be brought to MDPI's writing requirements (also SOME VALUABLE REFERENCES HAVE BEEN FORGOTTEN)!

The EXPERIMENTAL section should be more concise (the level at which it is presented is too low, many aspects are covered in specialized literature).

The figures must be more attractive or readable (possibly multicolored). For example figures 1, 4, 6 and 8.

The Conclusion section need a little polish.

Reviewer 3 Report

The paper presents a series of information about effect of kaolinite and cloisite Na+ on storage stability of rubberized binders

The paper may be of interest to the scientific community through the topic addressed.

Authors should consider the following observations:

- Avoid abbreviations in the abstract and keywords

- The introduction needs to be substantially improved. Other bibliographic sources relevant to the field should also be considered. Also, relationship 1 is not of any importance for the research thus carried out, so it should be eliminated.

- Figures 2 to 3 must be removed and replaced with macroscopic images of the samples used in the research;

- Pay close attention to the figures because they have an inadequate resolution;

- Perhaps other experimental tests should be carried out to highlight the advantages that the manufactured materials present in operation;

- The results obtained must be discussed in more detail to highlight their novelty in relation to other research in the field. In the form presented, a detailed analysis of the obtained results is not carried out. Thus, these results must be compared with other results obtained in other topical research in order to highlight the novelty of the research carried out. A scientific explanation of the results obtained in the research should also be presented. Simply finding that a parameter has increased or decreased is not enough;

- The conclusions are too general and repeat certain information from the abstract.

Round 2

Reviewer 1 Report

The revised manuscript can be accepted for publication

The revised manuscript can be accepted for publication

Reviewer 3 Report

The authors responded to the comments made and improved the paper accordingly.